# Efficacy of a mixture of Ginkgo biloba, sesame, and turmeric on cognitive function in healthy adults: Study protocol for a randomized, double-blind, placebo-controlled trial

**Taizen Nakase** [1,2]*, **Yasuko Tatewaki**[1], **Izumi Matsudaira**[2], **Kouki Kobayashi**[2], **Hikari Iki**[2], **Haruka Asaoka**[2], **Radiztia Ekayantri**[2], **Michiho Muranaka**[2], **Hiroyuki Murata**[3], **Tatsushi Mutoh**[2], **Yasuyuki Taki**[1,2]

1 Smart Aging Research Center, Tohoku University, Sendai, Japan, 2 Department of Aging Research and Geriatric Medicine, Institute of Development, Aging and Cancer, Tohoku University, Sendai, Japan, 3 Tohoku University Knowledge Cast Company Ltd., Sendai, Japan

* taizen.nakase.a4@tohoku.ac.jp

**Data Availability Statement:** No datasets were generated or analysed during the current study. All

## Abstract

### Background and purpose

Ginkgo biloba extract (GBE) reportedly ameliorates cognitive function in patients with chronic cerebrovascular insufficiency. However, its efficacy in healthy adults is ambiguous. It was reported that concentrations of terpene lactones, active components of GBE that are present in very low concentrations in the brain, were significantly increased following administration of a mixture of GBE, sesame seed, and turmeric (GBE/MST) in mice. This study aims to investigate the effectiveness of GBE/MST on the cognitive function of healthy adults by comparing it with that of GBE alone.

### Methods

Altogether, 159 participants providing informed consent will be recruited from a population of healthy adults aged 20–64 years. Normal cognitive function at baseline will be confirmed using the Japanese version of the Montreal Cognitive Assessment battery. Participants will be randomly assigned in a double-blind manner to the GBE/MST, GBE, and placebo groups in a 1:1:1 ratio. The Wechsler Memory Scale, Trail Making Test, and Stroop Color and Word Test will be used to assess the memory and executive functions at baseline and at the endpoint (24 weeks). For biological assessment, resting state functional magnetic resonance imaging (rs-fMRI) will be performed simultaneously with the neuropsychological tests.

### Discussion

This study aims to obtain data that can help compare the profile changes in memory and executive functions among participants consuming GBE/MST, GBE alone, and placebo for 24 weeks. Alterations in the default mode network will be evaluated by comparing the rs-

relevant data from this study will be made available upon study completion.

**Funding:** This study is a joint research with Ohki Pharmaceutical Company and funding is provided by the same company. GBE/MST supplement, GBE supplement, and placebo will also be provided by this company. The study protocol independently includes researchers at the Smart Aging Research Center and the acquired data cannot be accessed from outside this institute. The funder does not have any additional role in the study design, data collection and analysis, decision to publish, or preparation of the manuscript.

**Competing interests:** TN is paid a part of salary by the founder. This does not alter our adherence to PLOS ONE policies on sharing data and materials. YTat, IM, KK, HI, HA, RE, MM, TM, and YTak do not have any COI.

fMRI findings between baseline and 24 weeks in the aforementioned groups. Our results may clarify the impact of GBE on cognitive function and the functional mechanism behind altered cognitive function induced by GBE components.

## Trial registration

This study was registered in the University Hospital Medical Information Network Clinical Trials Registry (UMIN-CTR; registration number: UMIN000043494). This information can be searched on the website of the International Clinical Trials Registry Platform Search Portal of the World Health Organization under the Japan Primary Registries Network.

## Introduction

Complementary and alternative medicines, such as medical plants and functional foods, have been paid attention to as medication of various diseases in which the influence of lifestyle is pointed out [1–3]. In this context, it is desired to confirm the effect of these alternative agents by rational scientific evidences. In many European countries, GBE is available as a medicine for patients with chronic cerebrovascular insufficiency to reduce dizziness and improve memory. According to a meta-analysis involving nine clinical studies [4], a significant improvement was observed in the Syndrome-Kurztest scores following GBE administration in patients with Alzheimer's disease. There was no significant difference in Alzheimer's Disease Assessment Scale-Cognitive Subscale scores between patients consuming GBE and those consuming placebo. However, this difference was greater among patients consuming GBE in a dose of 240 mg/day compared to those on a dose of 120 mg/day. The difference was also greater in the observation period of 22–24 weeks compared to a period of 12 weeks. There was no difference in the adverse incident rate between GBE dose of 120 mg/day and 240 mg/day. Moreover, there was no difference in the dropout rate due to adverse effects between GBE doses of 120–160 mg/day or 240 mg/day and placebo [4, 5]. On the other hand, almost all studies were unable to show any efficacy of GBE in preventing Alzheimer's disease due to the low prevalence rate of Alzheimer's disease or low adherence rate to the study protocols [5].

GBE is available in the form of health supplements in Japan and the USA, with claims of ameliorating memory and attention and mitigating the symptoms of Alzheimer's disease. However, its efficacy in the cognitive function of healthy adults remains controversial. No effect was reported on memory enhancement and on the intelligence test scores after 6 days of GBE administration in healthy adults aged 20 years [6]. Reportedly, healthy elderly individuals showed no improvement in memory but showed improvements in executive functions after 6 weeks of GBE administration [7]. There was no difference in the prevalence rate of cognitive decline between the GBE and placebo groups for 42 months among healthy adults aged over 85 years. However, among participants who maintained good adherence throughout the study period, GBE was associated with a significantly reduced rate of cognitive decline when compared with placebo [8].

Originally, GBE has been reported to show anti-platelet effect, anti-inflammatory effect, and inhibiting effect of amyloid-β aggregation [9–11]. From the view point of functional component, two active elements can be found in GBE, i.e. flavonoids and terpene lactones. Flavonoids have been reported to accumulate in the blood serum [12]. Since major effects of flavonoids include reduction in the inflammatory response, oxidization, and platelet aggregation in vivo, these effects are expected to play important roles in inhibiting damage to the

vascular endothelial cells and improving the peripheral blood flow [13]. Terpene lactones have been reported to infiltrate the central nervous system (CNS) but are quickly metabolized, resulting in very low concentrations in the CNS [12]. Since terpene lactones were reported to reduce caspase-3 activation and amyloid-β aggregation and protect the mitochondrial membrane, they can be expected to protect the neurons from apoptosis and amyloid-β toxicity [14]. However, since the concentration of terpene lactones in the CNS might be very low in vivo [12, 15], the effect of GBE on cognitive function may be exclusively explained by improvement in the cerebral blood flow. Administration of a mixture of GBE, sesame seed, and turmeric (GBE/MST) was reported to increase the concentration of terpene lactones in the mouse brain [15]. The aforementioned study reported that the concentration of bilobalide, a terpene lactone, in the brain was not significantly different between the GBE alone and GBE/MST groups (mean ± standard error of the mean: 1.30±0.31 and 0.75±0.24 μg/mg, respectively). Ginkgolide A, another terpene lactone, was detected in the GBE/MST group, but it was not detected in the GBE alone group. The GBE/MST supplement is currently available in the Japanese market, although its scientific effects have not been validated. Since low bioavailability of GBE in the brain may be a possible cause of its faint effect on the cognitive function, it might be of great interest to investigate whether the efficacy of GBE is improved by administration in the form of GBE/MST in healthy adults.

## Materials and methods

### Study design

This 24-week, randomized, double-blind, placebo-controlled study aims to discriminate the effect of GBE/MST, and the group in which the original GBE supplement was provided was established as a counter part of the GBE/MST group. Local residents will be recruited through advertisements in local newspapers and flyers, and they will be divided into three groups (GBE/MST, GBE, and placebo groups). Eligibility will be screened by a principal investigator.

The primary aim of this study is to determine the efficacy of GBE/MST in the impact of cognitive function when compared with that of GBE alone and placebo in healthy adults using neuropsychological assessments. The study also aims to evaluate the alterations in the default mode network (DMN) after 24 weeks of GBE/MST administration and compare them with the baseline data. SPIRIT Table is shown in Fig 1, and a flowchart of the study protocol is shown in Fig 2.

Inclusion criteria are 1) male and female participants aged between 20 and 64 years and 2) normal cognitive function at baseline, confirmed by the Japanese version of the Montreal Cognitive Assessment battery (MoCA-J) (score ≧26). Exclusion criteria are 1) history of severe neurological diseases, 2) history of critical bleeding incidents, 3) use of antithrombotic medications, 4) severe depression, 5) diabetes mellitus, 6) severe visual or auditory disability, 7) metal implantation such as cardiac pacemaker, coronary stent, and artificial joints, 8) claustrophobia, 9) use of any supplements, 10) creatinine clearance below 15 mL/min, 11) gastrointestinal diseases or metabolic diseases, 12) diagnosed psychiatric disorders, 13) participants who cannot provide informed consent, 14) allergy to the study materials, and 15) individuals not suitable for participation based on the decision of the principal investigator.

### Sample size

The primary endpoint of this study is to determine the impact of GBE/MST on cognitive function in healthy adults, which will be assessed by the Wechsler Memory Scale-Revised (WMS-R) after 24 weeks of GBE/MST administration. The WMS-R is designed to indicate a standard deviation of 15 if its normal average score is 100. In the present study, when the

| | STUDY PERIOD | | | |
|---|---|---|---|---|
| | Enrolment | Allocation | Post-allocation | Close-out |
| **TIMEPOINT** | *0* | *0* | *24±2 w* | |
| **ENROLMENT:** | | | | |
| **Eligibility screen** | X | | | |
| **Informed consent** | X | | | |
| **Allocation** | | X | | |
| **INTERVENTIONS:** | | | | |
| *GBE/MST* | | | ●———● | |
| *GBE* | | | ●———● | |
| *Placebo* | | | ●———● | |
| **ASSESSMENTS:** | | | | |
| *MoCA-J* | X | | | |
| *Blood test* | X | | X | X |
| *MRI* | X | | X | X |
| *Cognitive tests** | X | | X | |

**Fig 1. SPIRIT table.** *Cognitive tests include WMS-R, Stroop color & word test, and Trail making test. GBE: ginkgo biloba extract, GBE/MST: mixture of GBE, sesame seed, and turmeric, MoCA-J: Japanese version of Montreal cognitive assessment battery, MRI: magnetic resonance imaging.

estimated alteration rate of the standard deviation was 1%, the difference in the average score was 6. Cohen's calculation formula with an effect size of 0.4 at a significance level ($\alpha$) of 0.05 and a detection power (1-$\beta$) of 80% yielded a sample size of 44 participants. Considering a dropout rate of 20%, 159 participants will be required (53 participants in each group).

## Intervention

**Active.** GBE/MST capsule (commercially known as "Brain assist ®") contains GBE, sesame seed, turmeric, and excipients (Table 1), whereas the GBE capsule (commercially known as "Icho-ba Ekisu") contains GBE and excipients (Table 1). Both have a dosage of two capsules per day, that is, each participant will take 120mg of GBE per day. Both the products will have similar shapes, colors, and tastes. The package containing the products will not have any information to avoid participant bias.

**Placebo.** Placebo will be made in matched for size, color and taste to the GBE/MST capsule and GBE capsule. The precise ingredients can be found in Table 1.

Therefore, participants in the GBE/MST group will consume two capsules of the GBE/MST supplement. Participants in the GBE group will consume two capsules of GBE supplement. Participants in the placebo group will consume two capsules of placebo.

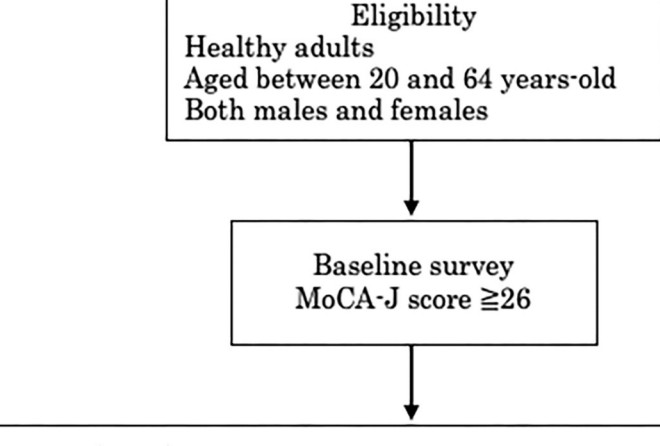

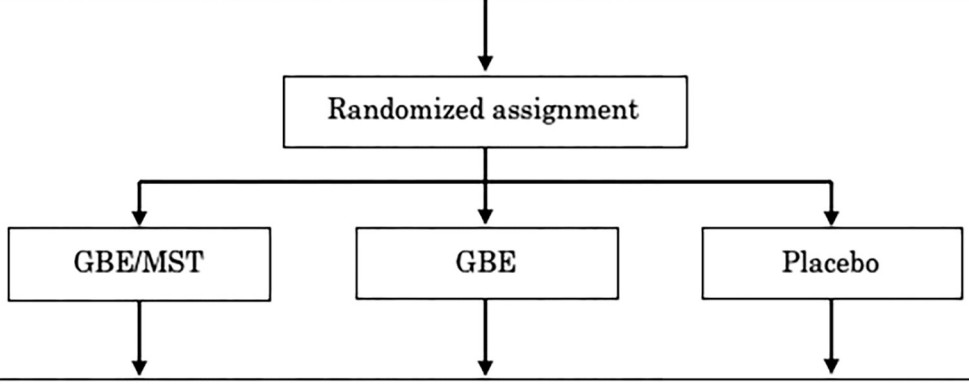

**Fig 2. Flowchart of the study protocol.** GBE/MST: A mixture of Ginkgo biloba extract, sesame seed, and turmeric; GBE: Ginkgo biloba extract; MRI: magnetic resonance imaging; rs-fMRI: resting state functional magnetic resonance imaging.

 **Randomization.** Participants will be randomly divided into the GBE/MST, GBE, and placebo groups in a 1:1:1 ratio, by a random number generating method with computer assistance. An operator does not participate in the examination. For maintaining a double-blind manner, both participants and examiners cannot obtain any information about the assigned agent throughout the study period.

**Table 1. Precise data of active and control products.**

| Name | GBE/MST | GBE | Placebo |
|---|---|---|---|
| Shape | Capsule | Capsule | Capsule |
| Ingredients | Gingko biloba extract, sesame seed, turmeric oil, reduced sugar syrup, refined fish oil containing docosahexaenoic acid with gelatin, spice extract, glycerin, glycerin fatty acid ester, beeswax, caramel color, vegetable lecithin from soybean | Gingko biloba extract, reduced sugar syrup, gelatin, glycerin fatty acid ester, caramel color, tocotrienol lecithin from soybean | Reduced sugar syrup, refined fish oil containing docosahexaenoic acid with gelatin, spice extract, glycerin, glycerin fatty acid ester, beeswax, caramel color, vegetable lecithin from soybean |
| Energy | 4 kcal, proteins: 0.2g, lipids: 0.3g, carbohydrates: 0.2g, salt: 0.0008g | 5 kcal, proteins: 0.2g, lipids: 0.4g, carbohydrates: 0.2g, salt: 0.0009g | 5 kcal, proteins: 0.2g, lipids: 0.5g, carbohydrates: 0.1g, salt: 0.0009g |

**Adverse effects and cessation criteria.** The adverse effects will be screened throughout the study period. Headache, dizziness, tinnitus, diarrhea, nausea, gastrointestinal symptoms, high blood pressure, excessive thirst, dyspnea, chest pain, bradycardia, and upper respiratory infection have been reported as adverse effects of GBE [5]. Interestingly, participants consuming GBE showed a lower incidence of adverse events when compared with those consuming a placebo [16]. The study will be terminated if any grade 3 adverse event is reported. If a participant shows any adverse events and requests a refusal, the study will be terminated. If a principal investigator decides a participant will not follow the protocol, the study will be terminated.

**Testing schedule.** At registration, written informed consent will be obtained from the participants. Subsequently, cognitive function will be evaluated using the MoCA-J. Participants with a score of $\geqq 26$ will be eligible for inclusion.

Standard blood and urine tests will be performed to check for health status. Body measurements, grasping power of both the hands, two-step test, and functional reach test will be performed at baseline and at the endpoint (24 weeks).

Cognitive tests will be completed by the participants at baseline and at the endpoint. The WMS-R and its lower domains will be used to measure total memory function and assess individual function (digit span indicates verbal memory and the tapping span indicates visual memory). Stroop Color and Word Test and Trail Making Test will be employed to assess executive functions such as restraint-related functions and sustained attention, respectively.

Magnetic resonance imaging (MRI: Achieva Intera 3T, Philips Healthcare, Quasar Dual, The Netherlands) will be performed at baseline and at the endpoint. T1 weighted magnetization-prepared rapid acquisition gradient echo (MPRAGE) structural images (matrix: $240 \times 240$, repetition time: 8.70 ms, echo time: 3.1 ms, flip angle: 8˚, field of view: $256 \times 256 \times 180$ mm, 162 slices, voxel size: $0.7 \times 0.7 \times 0.7$ mm, and scan duration: 5 min 15 s) will be acquired to collect 3D structural datasets. Resting state functional MRI (rs-fMRI: $T2^*$-weighted gradient-echo planar imaging, matrix: $64 \times 64$, repetition time: 3000 ms, echo time: 30 ms, flip angle: 90˚, field of view: $220 \times 220 \times 218$ mm, 34 slices, voxel size: $3.44 \times 3.44 \times 3.40$ mm and 197 volumes with 5 min superimposition) with closed eyes will be performed to analyze the DMN.

**Strategies to improve adherence to intervention protocols.** For improving adherence to the study protocol, all participants are required to check a calendar for taking the capsules. Participants are also asked to bring back any remaining capsules at the endpoint visit. For preventing any unknown effects, participants are prohibited to take any additional supplements nor drugs during study period.

**Statistical analysis.** The average of total WMS-R score at the endpoint will be compared among the GBE/MST, GBE, and placebo groups using one-way analysis of variance (ANOVA) and Tukey's test. The difference between the total score at baseline and at the endpoint will be compared among the three groups using multivariate ANOVA (MANOVA). The average

values of the forward and backward digit spans will be compared among the three groups using ANOVA and Tukey's test. The average values of forward and backward tapping spans will be compared in the same manner as that used for the digit span results. In addition, the difference between baseline values and those at the endpoint will be evaluated among the three groups using MANOVA. The average time (second) and average count of misses in the Stroop Color and Word Test at the endpoint will be compared among the three groups using ANOVA and Tukey's test. The differences in time and count between baseline and the endpoint will also be analyzed among the three groups using MANOVA. Data from the Trail Making Test will be evaluated in the same way as those form Stroop Color and Word Test.

Total cerebral cortical volume will be calculated from the T1-weighted structural images using the SPM12 software (statistical parametric mapping: https://www.fil.ion.ucl.ac.uk/spm/software/spm12/). The average alteration in total volume from baseline to the endpoint will be compared among the three groups. Using the CONN toolbox (https://web.conn-toolbox.org/) implemented in MATLAB, functional connectivity (FC) between two regions of interest (ROIs) within the DMN will be analyzed by means of ROI-to-ROI analysis. The differences in FC will be compared among the three groups.

All statistical analyses will be performed using JMP Pro15 (JMP Inc., Cary, NC, USA).

**Ethical issue.** This study has been registered in the University Hospital Medical Information Network Clinical Trials Registry (UMIN-CTR) (registration number: UMIN000043494). This information can be searched on the website of the International Clinical Trials Registry Platform Search Portal of the World Health Organization, under the Japan Primary Registries Network. All protocols used in this study have been approved by the Ethics Committee of Tohoku University Graduate School of Medicine (#2021-1-039). All participants will be informed about the study using a written informed consent form, which has been confirmed by the Institutional Ethical Committee. Personal data of all participants will be collected and stored with strictly protected instruments at our facility. Personal data will be anonymized before the statistical analysis. The principal investigator will be responsible for all the data.

When changing the research plan, the research director will obtain the approval of the head of a research institution after review by the Ethics Committee. The data obtained in this study are expected to be used in another or future researches approved by the Ethics Committee. Confirm and follow the intention at the time of participation in the research on whether or not the results can be diverted. If the research to be diverted has not already been approved by the Ethics Committee, apply again to the Ethics Committee.

**Dissemination.** The findings will be presented at relevant research conferences, seminars, and academic meetings. They will also be published in a peer-reviewed international journal from the cognitive sciences sector.

## Discussion

GBE has been reported to have two major active components: flavonoids and terpene lactones. Flavonoids remain in the circulating plasma, maintaining endothelial cells via antiplatelet and anti-inflammatory effects [13]. Terpene lactones are transferred to the brain and play an important role in inhibiting caspase-3 activity and amyloid-β aggregation, thus protecting the neurons [14]. However, most of the terpene lactones are immediately metabolized, leading to very low concentrations in the brain. Therefore, the efficacy of GBE on cognitive function in healthy adults can be explained mainly by the improvement in the cerebral blood flow while maintaining the endothelial cells. Recently, our colleagues reported that the density of terpene lactones in the mouse brain was specifically increased following administration of MST [15]. Therefore, if participants consuming GBE/MST show any additional alterations in cognitive

function when compared to those consuming GBE alone, the direct effect of terpene lactones on brain function will become remarkable. Moreover, by comparing the affected domain of cognitive function with alterations in the DMN following GBE/MST administration, we can provide precise information about the influence of GBE components on the brain. Thus, the present study has the potential to elucidate the mechanism behind the stimulation of cognitive function induced by GBE components.

## Supporting information

**S1 Checklist. SPIRIT 2013 checklist: Recommended items to address in a clinical trial protocol and related documents**∗.
(DOC)

**S1 File.**
(DOCX)

**S2 File.**
(DOCX)

## Acknowledgments

We are grateful to Mr. Hirokazu Kawamoto and Mr. Fumiaki Takeshita for their support of preparing study materials. We also thank Ms. Sayaka Makabe, Ms. Misaki Abe, Ms. Maiko Chiba, and Ms. Chieko Miura for their help with the study preparation. We would like to thank Editage (www.editage.com) for English language editing.

## Author Contributions

**Conceptualization:** Taizen Nakase, Hiroyuki Murata, Yasuyuki Taki.

**Data curation:** Taizen Nakase, Yasuko Tatewaki, Izumi Matsudaira, Kouki Kobayashi, Hikari Iki, Haruka Asaoka, Radiztia Ekayantri, Michiho Muranaka.

**Formal analysis:** Taizen Nakase, Yasuko Tatewaki.

**Funding acquisition:** Hiroyuki Murata, Yasuyuki Taki.

**Investigation:** Taizen Nakase, Kouki Kobayashi, Hikari Iki, Haruka Asaoka, Radiztia Ekayantri, Michiho Muranaka, Tatsushi Mutoh, Yasuyuki Taki.

**Methodology:** Taizen Nakase, Yasuko Tatewaki, Izumi Matsudaira, Kouki Kobayashi, Haruka Asaoka, Tatsushi Mutoh.

**Project administration:** Yasuyuki Taki.

**Supervision:** Tatsushi Mutoh, Yasuyuki Taki.

**Validation:** Tatsushi Mutoh.

**Writing – original draft:** Taizen Nakase.

**Writing – review & editing:** Yasuyuki Taki.

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
