## [Decision Letter · Decision Letter 0]

6 Sep 2022

PONE-D-22-03597Efficacy of a mixture of Ginkgo biloba extract, sesame extract, and turmeric oil on cognitive function in healthy adults: protocol for a 24-week, randomized, double-blind, placebo-controlled trialPLOS ONE

Dear Dr. Nakase,

Thank you for submitting your manuscript to PLOS ONE. After careful consideration, we feel that it has merit but does not fully meet PLOS ONE’s publication criteria as it currently stands. Therefore, we invite you to submit a revised version of the manuscript that addresses the points raised during the review process.

The manuscript has been evaluated by two reviewers, and their comments are available below.

One of the reviewers is satisfied with your submission, but the other has a number of suggestions for changes.

I agree with reviewer 2 that the inclusion and exclusion criteria can be presented as text, rather than as bullet points, and that the quality figure 1 needs improvement. I also agree that further details on blinding and randomization could be provided. However, I will leave it up to you whether to make changes to your title and keywords, and also whether you think it is appropriate to include any discussion of CAM treatments for various diseases.

Please also see my additional requests below regarding the SPIRIT schedule and checklist.

We look forward to receiving your revised manuscript.

Kind regards,

Steve Zimmerman, PhD

Associate Editor, PLOS ONE

Journal Requirements:

3. Thank you for providing the following Funding Statement: 

“I have read the journal's policy and the authors of this manuscript have the following competing interests: HK is employee of Ohki Pharmaceutical Company.”

We note that one or more of the authors is affiliated with the funding organization, indicating the funder may have had some role in the design, data collection, analysis or preparation of your manuscript for publication; in other words, the funder played an indirect role through the participation of the co-authors.

If the funding organization did not play a role in the study design, data collection and analysis, decision to publish, or preparation of the manuscript and only provided financial support in the form of authors' salaries and/or research materials, please review your statements relating to the author contributions, and ensure you have specifically and accurately indicated the role(s) that these authors had in your study in the Author Contributions section of the online submission form. Please make any necessary amendments directly within this section of the online submission form.  Please also update your Funding Statement to include the following statement: “The funder provided support in the form of salaries for authors [insert relevant initials], but did not have any additional role in the study design, data collection and analysis, decision to publish, or preparation of the manuscript. The specific roles of these authors are articulated in the ‘author contributions’ section.”

If the funding organization did have an additional role, please state and explain that role within your Funding Statement.

Additional Editor Comments (if provided):

SPIRIT schedule is missing:

Please note that you must upload a completed SPIRIT schedule as Fig 1 of your manuscript. Blank copies of this document and information regarding SPIRIT schedules can be found here: http://www.spirit-statement.org/schedule-of-enrolment-interventions-and-assessments/

Completed SPIRIT checklist is missing:

Our author guidelines for Clinical Trial Study Protocols require that you upload a SPIRIT checklist as Supporting Information. Information about the SPIRIT guidance and blank checklists can be found here: https://www.spirit-statement.org/.

More information related to the relevant journal policy is available here: https://journals.plos.org/plosone/s/submission-guidelines#loc-study-protocols

Reviewers' comments:

Reviewer's Responses to Questions

**Comments to the Author**

1. Does the manuscript provide a valid rationale for the proposed study, with clearly identified and justified research questions?

Reviewer #1: Yes

Reviewer #2: Yes

2. Is the protocol technically sound and planned in a manner that will lead to a meaningful outcome and allow testing the stated hypotheses?

Reviewer #1: Yes

Reviewer #2: Yes

3. Is the methodology feasible and described in sufficient detail to allow the work to be replicable?

Reviewer #1: Yes

Reviewer #2: Yes

4. Have the authors described where all data underlying the findings will be made available when the study is complete?

Reviewer #1: Yes

Reviewer #2: Yes

5. Is the manuscript presented in an intelligible fashion and written in standard English?

Reviewer #1: Yes

Reviewer #2: Yes

6. Review Comments to the Author

You may also provide optional suggestions and comments to authors that they might find helpful in planning their study.

Reviewer #1: This protocol provide a valid rational for a proposed study. It describes with sufficient details the planned study. The study is planned as a double blind, placebo controlled study. Participants are well described (included after informed consent), randomly assigned to GBE/MST, GBE or placebo 1:1:1. Inclusion and exclusion criteria are described with details. Planned psychological tests are informative and give sufficient data on cognitive functions. Resting state MRI would be done simultaneously with psychological testing.

Statistical methods are eligible and well described. Altogether, methodology is described with sufficient details to allow the work to be replicable.

The authors described their plans for enabling the results of this study to be visible: presenting at scientific and professional gatherings and publishing in peer-reviewed journals..

The manuscript is presented in an intelligible fashion and written in standard English.

The planned study contributes to clarifying the impact of GBE alone and of GBE/MST on cognitive functions and functional mechanisms behind.

Reviewer #2: 1. It is suggested to change your title from “Efficacy of a mixture of Ginkgo biloba extract, sesame extract, and turmeric oil on cognitive function in healthy adults: protocol for a 24-week, randomized, double-blind, placebo-controlled trial” to “Efficacy of a mixture of Ginkgo biloba, sesame, and turmeric on cognitive function in healthy adults: study protocol for a randomized, double-blind, placebo-controlled triall.

2. “Integrative medicine” and “phytotherapy” should be added to the keywords. “Ginkgo biloba extract; sesame extract; turmeric oil” should be changed to “Ginkgo biloba; sesame; turmeric”.

3. At the beginning of the introduction, a paragraph on CAM position for treatment of various diseases and the necessity for an evidence-based approach in CAM should be added. The below-mentioned references are strongly recommended to be cited there:

- Paudyal, Vibhu, et al. "Complementary and alternative medicines use in COVID-19: A global perspective on practice, policy and research." Research in Social and Administrative Pharmacy 18.3 (2022): 2524-2528.

- Hashemi, Monire Seyed, et al. "Efficacy of pomegranate seed powder on glucose and lipid metabolism in patients with type 2 diabetes: a prospective randomized double-blind placebo-controlled clinical trial." Complementary Medicine Research 28.3 (2021): 226-233.

- Wu, Jinhui, et al. "Complementary and Alternative Medicine Use by ADHD Patients: A Systematic Review." Journal of Attention Disorders (2022): 10870547221111557.

4. The inclusion and exclusion criteria should be re-written as text (not bullet points).

5. In Table 1, “Piper longum extract” has been written instead of “turmeric oil”. What about its discrepancies? In addition, it is suggested to use common name or scientific name, through the manuscript.

6. Blinding should be described in detail. Also, details of randomization should be expanded.

7. Figure 1 has not acceptable quality.

7. PLOS authors have the option to publish the peer review history of their article (what does this mean?). If published, this will include your full peer review and any attached files.

Reviewer #1: **Yes: **Vida Demarin

Reviewer #2: No

---

## [Author Response · Author response to Decision Letter 0]

29 Sep 2022

Response to editor and reviewers.

> I agree with reviewer 2 that the inclusion and exclusion criteria can be presented as text, rather than as bullet points, and that the quality figure 1 needs improvement. I also agree that further details on blinding and randomization could be provided. 

 Thank you very much for your effective comments. We rewrote the inclusion and exclusion criteria in text style (p5 line19 – line29). Figure 2, previous Fig 1, was corrected the quality. We described the section of Randomization more precisely (p7 line8 – line13). 

> However, I will leave it up to you whether to make changes to your title and keywords, and also whether you think it is appropriate to include any discussion of CAM treatments for various diseases.

 Thank you very much for your suggestion. We changed the title following comment of Reviewer #2 (p1 line1 – line2). However, because our study is not for medical treatment, we did not include keywords, “integrative medicine” and “phytotherapy”. At the begging of the introduction section, we briefly mentioned the meaning of CAM following Reviewer #2’s comment (p3 line2 – line5). 

>2. We note that the grant information you provided in the ‘Funding Information’ and ‘Financial Disclosure’ sections do not match.

 We corrected and matched the funding information and financial disclosure (p1 line13 – line23).

>3. Thank you for providing the following Funding Statement: “I have read the journal's policy and the authors of this manuscript have the following competing interests: HK is employee of Ohki Pharmaceutical Company.”

We note that one or more of the authors is affiliated with the funding organization, indicating the funder may have had some role in the design, data collection, analysis or preparation of your manuscript for publication; in other words, the funder played an indirect role through the participation of the co-authors.

If the funding organization did not play a role in the study design, data collection and analysis, decision to publish, or preparation of the manuscript and only provided financial support in the form of authors' salaries and/or research materials, please review your statements relating to the author contributions, and ensure you have specifically and accurately indicated the role(s) that these authors had in your study in the Author Contributions section of the online submission form. Please make any necessary amendments directly within this section of the online submission form. Please also update your Funding Statement to include the following statement: “The funder provided support in the form of salaries for authors [insert relevant initials], but did not have any additional role in the study design, data collection and analysis, decision to publish, or preparation of the manuscript. The specific roles of these authors are articulated in the ‘author contributions’ section.”

If the funding organization did have an additional role, please state and explain that role within your Funding Statement.

 Actually, HK and FT do not participate in execution of the study. As they will only provide study materials, we removed their name from authors list and put their name in the acknowledge section. They, of course, agreed our decision. We rewrote the COI statement (p1 line4 – line5, p11 line15 – line17). Additionally, one listed author, YZ, has not been contributing to this study anymore. So that we deleted this name from authors list.

>4. Your ethics statement should only appear in the Methods section of your manuscript. If your ethics statement is written in any section besides the Methods, please delete it from any other section.

 Following your comment, we deleted the ethics statement from abstract (p2 line10). We included the ethics statement only in the methods section (p9 line25 – p10 line11).

>5. Please review your reference list to ensure that it is complete and correct. If you have cited papers that have been retracted, please include the rationale for doing so in the manuscript text, or remove these references and replace them with relevant current references. Any changes to the reference list should be mentioned in the rebuttal letter that accompanies your revised manuscript. If you need to cite a retracted article, indicate the article’s retracted status in the References list and also include a citation and full reference for the retraction notice.

 Following Reviewer #2’s comment, we wrote the status of CAM and cited additional references which support our discussion (p3 line2 – line5).

>SPIRIT schedule is missing:

 We included SPIRIT figure in the manuscript as Figure 1. (p5 line8 – line14)

>Completed SPIRIT checklist is missing:

 We completed SPIRIT checklist and uploaded. Moreover, following SPIRIT lists, we newly described additional descriptions in the text (p5 line13, p6 line16, p7 line8 – line13, p7 line22 – line23, p8 line24 – line29, p10 line5 – line11).

 Throughout text, we changed some words and sentences for keeping consistency of the manuscript. All changed parts were highlighted in red. 

To Reviewer #1,

Thank you very much. We appreciate to your encouraging comments.

To Reviewer #2, 

>1. It is suggested to change your title from “Efficacy of a mixture of Ginkgo biloba extract, sesame extract, and turmeric oil on cognitive function in healthy adults: protocol for a 24-week, randomized, double-blind, placebo-controlled trial” to “Efficacy of a mixture of Ginkgo biloba, sesame, and turmeric on cognitive function in healthy adults: study protocol for a randomized, double-blind, placebo-controlled triall.

 Thank you very much for your effective suggestion. Now, we changed the words of title following your comment (p1 line1 – line2). 

>2. “Integrative medicine” and “phytotherapy” should be added to the keywords. “Ginkgo biloba extract; sesame extract; turmeric oil” should be changed to “Ginkgo biloba; sesame; turmeric”.

 Thank you for your comments. We corrected the material names in the keywords following your comment (p2 line30 – line31). However, because our study is focusing not on the medical effect of GBE but on the influence of GBE on the cognitive function of healthy adults, we did not include these two keywords you suggested. 

>3. At the beginning of the introduction, a paragraph on CAM position for treatment of various diseases and the necessity for an evidence-based approach in CAM should be added. The below-mentioned references are strongly recommended to be cited there:

- Paudyal, Vibhu, et al. "Complementary and alternative medicines use in COVID-19: A global perspective on practice, policy and research." Research in Social and Administrative Pharmacy 18.3 (2022): 2524-2528.

- Hashemi, Monire Seyed, et al. "Efficacy of pomegranate seed powder on glucose and lipid metabolism in patients with type 2 diabetes: a prospective randomized double-blind placebo-controlled clinical trial." Complementary Medicine Research 28.3 (2021): 226-233.

- Wu, Jinhui, et al. "Complementary and Alternative Medicine Use by ADHD Patients: A Systematic Review." Journal of Attention Disorders (2022): 10870547221111557.

 Thank you for your suggestion. We mentioned the current situation of CAM at the beginning of introduction section including references as you suggested (p3 line2 – line5).

>4. The inclusion and exclusion criteria should be re-written as text (not bullet points).

 Thank you for your suggestion. We rewrote the inclusion and exclusion criteria in text style (p5 line19 – line29).

>5. In Table 1, “Piper longum extract” has been written instead of “turmeric oil”. What about its discrepancies? In addition, it is suggested to use common name or scientific name, through the manuscript.

 Thank you for your careful reading. That words were our mistake. We corrected the name of the material (p6 line4 – 5 in Table 1).

>6. Blinding should be described in detail. Also, details of randomization should be expanded.

 Thank you for your comment. We described the blinding procedure and the method of randomization more precisely (p7 line8 – line13).

>7. Figure 1 has not acceptable quality.

 We rewrote Figure 2 (former Figure 1), more carefully.

---

## [Decision Letter · Decision Letter 1]

4 Jan 2023

Efficacy of a mixture of Ginkgo biloba, sesame, and turmeric on cognitive function in healthy adults: study protocol for a randomized, double-blind, placebo-controlled trial

PONE-D-22-03597R1

Dear Dr.  Nakase,

We’re pleased to inform you that your manuscript has been judged scientifically suitable for publication and will be formally accepted for publication once it meets all outstanding technical requirements.

Please amend the competing interests statement to clarify whether the funder had any role in the study design and analysis as specified in the comments.

Kind regards,

Anya Topiwala

Academic Editor

PLOS ONE

Additional Editor Comments (optional):

The authors have done a good job of addressing all the remaining comments from the reviewers. I could only find one omission - the competing interests statement only states that one authors has salary paid by the funder but does not specify if the funder had any additional role in the study.

Reviewers' comments:

Reviewer's Responses to Questions

**Comments to the Author**

1. Does the manuscript provide a valid rationale for the proposed study, with clearly identified and justified research questions?

Reviewer #1: Yes

Reviewer #2: Yes

2. Is the protocol technically sound and planned in a manner that will lead to a meaningful outcome and allow testing the stated hypotheses?

Reviewer #1: Yes

Reviewer #2: Yes

3. Is the methodology feasible and described in sufficient detail to allow the work to be replicable?

Reviewer #1: Yes

Reviewer #2: No

4. Have the authors described where all data underlying the findings will be made available when the study is complete?

Reviewer #1: Yes

Reviewer #2: Yes

5. Is the manuscript presented in an intelligible fashion and written in standard English?

Reviewer #1: Yes

Reviewer #2: Yes

6. Review Comments to the Author

You may also provide optional suggestions and comments to authors that they might find helpful in planning their study.

Reviewer #1: Dear Authors,

This manuscript provide a valid rationale for the proposed study. The protocol is technically sound and planned in the manner to allow testing of your stated hypothesis. The work can be replicable as you described the methodology in sufficient details. The manuscript is well written and proposed study protocol could be used for randomized, double-blind, placebo-controlled trial.

Reviewer #2: 1. Reference No. 1 is not related to your work. It is related to the nutrition and food industry! My previous suggested reference (Hashemi, Monire Seyed, et al. "Efficacy of pomegranate seed powder on glucose and lipid metabolism in patients with type 2 diabetes: a prospective randomized double-blind placebo-controlled clinical trial." Complementary Medicine Research 28.3 (2021): 226-233.) should be added instead of it.

2. Regarding my previous comment (Blinding should be described in detail. Also, details of randomization should be expanded.) it is necessary to follow the CONSORT checklist. How did you guarantee the blinding process? For randomization, did you use blocked randomization, stratified randomization…?

7. PLOS authors have the option to publish the peer review history of their article (what does this mean?). If published, this will include your full peer review and any attached files.

Reviewer #1: No

Reviewer #2: No

---

## [Editor Report · Acceptance letter]

6 Mar 2023

PONE-D-22-03597R1 

Efficacy of a mixture of Ginkgo biloba, sesame, and turmeric on cognitive function in healthy adults: study protocol for a randomized, double-blind, placebo-controlled trial 

Dear Dr. Nakase:

I'm pleased to inform you that your manuscript has been deemed suitable for publication in PLOS ONE. Congratulations! Your manuscript is now with our production department. 

Kind regards, 

on behalf of

Dr. Anya Topiwala 

Academic Editor

PLOS ONE